# Challenges in Coopted Hydrophilic and Lipophilic Herbal Bioactives in the Same Nanostructured Carriers for Effective Bioavailability and Anti-Inflammatory Action

**DOI:** 10.3390/nano11113035

**Published:** 2021-11-12

**Authors:** Teodora-Alexandra Iordache, Nicoleta Badea, Mirela Mihaila, Simona Crisan, Anca Lucia Pop, Ioana Lacatusu

**Affiliations:** 1Faculty of Applied Chemistry and Materials Science, University POLITEHNICA of Bucharest, Polizu No 1, 011061 Bucharest, Romania; teodora.manasia@stud.chimie.upb.ro (T.-A.I.); nicoleta.badea@upb.ro (N.B.); 2National Research & Development Institute for Food Bioresources—IBA Bucharest, 6th Dinu Vintila Street, 021102 Bucharest, Romania; 3Virology Institute “Stefan S. Nicolau”, Romanian Academy, Mihai Bravu Street No 285, 030304 Bucharest, Romania; mirela.mihaila@virology.ro; 4RD Center, AC HELCOR, Victor Babes St., 430082 Baia Mare, Romania; simonacrisan@achelcor.ro; 5Faculty of Pharmacy, “Carol Davila” University of Medicine and Pharmacy, 6 Traian Vuia Street, 020945 Bucharest, Romania

**Keywords:** diosgenin-nanocarriers, wild yam extract, short-life ROS, cationic stable radicals

## Abstract

There is ongoing research on various herbal bioactives and delivery systems which indicates that both lipid nanocarriers and herbal medicines will be fine tunned and integrated for future bio-medical applications. The current study was undertaken to systematically develop NLC-DSG-yam extract for the improved efficacy of herbal Diosgenin (DSG) in the management of anti-inflammatory disorders. NLCs were characterized regarding the mean size of the particles, morphological characteristics, physical stability in time, thermal behaviour, and entrapment efficiency of the herbal bioactive. Encapsulation efficiency and in vitro antioxidant activity measured the differences between the individual and dual co-loaded-NLC, the co-loaded one assuring a prolonged controlled release of DSG and a more emphasized ability of capturing short-life reactive oxygen species (ROS). NLCs safety properties were monitored following the in vitro MTS ((3-(4,5-dimethylthiazol-2-yl)-2,5-diphenyltetrazolium bromide) tetrazolium reduction assay) and RTCA (Real-Time Cell Analysis) assays. Concentrations less than 50 μg/mL showed no cytotoxic effects during in vitro cytotoxicity assays. Besides, the NLC-DSG-yam extract revealed a great anti-inflammatory effect, as the production of pro-inflammatory cytokines (TNF-alpha, IL-6) was significantly inhibited at 50 μg/mL NLC (e.g., 98.2% ± 1.07 inhibition of TNF-α, while for IL-6 the inhibition percentage was of 62% ± 1.07). Concluding, using appropriate lipid nanocarriers, the most desirable properties of herbal bioactives could be improved.

## 1. Introduction

Herbal bioactives are gaining more interest as components of novel formulations, adding health beneficial properties due to their reduced toxicity compared with synthetic ingredients. Actual health trends lean more and more on natural resources. The bioactive phytochemicals from various plants have been used to treat health problems for a long time [1]. Many bioactive phytochemicals, such as polyphenols, flavonoids, steroids, terpenoids, and alkaloids from medicinal plants, have been used as precursors for several synthetic and semisynthetic drugs [2]. Administration of herbal drugs, however, is not always the most efficacious approach. The effectiveness of herbal bioactives depends on the active compound’s bioavailability over a sustained period. Unfortunately, most herbal bioactives manifest low bioavailability [3]. Moreover, fortified food or dietary supplements rich in herbal bioactives are hard enough to obtain and preserve because the bioactive substances are sensitive to light, temperature, moisture, or the internal gastrointestinal tract [4,5]. One viable solution is nanoencapsulation. Nanoencapsulation has gained increased interest in the food and pharmaceutical industries due to its advantages over existing technologies. It is a necessary and protective tool for entrapping sensitive bioactive compounds offering targeted control release, better oral availability, increased solubility and stability, as well as prolonged shelf life [6,7]. Hence, nanoencapsulation requested a delivery system, which was represented by an appropriate nanocarrier. According to the needs of the industry at the time, many types of nanocarrier have been developed, either based on polymers (synthetic or natural ones) or other nanocarriers, such as liposomes, microemulsions, lipid nanoparticles, and others [8,9]. These delivery systems have been employed to increase drug bioavailability, minimize drug degradation, and assure a sustained drug release. Among various delivery systems, nanostructured lipid carriers (NLC) are effective due to their biocompatibility, bioavailability, stability, as well as controlled and drug targeted abilities [10,11]. NLC consists in a mixture of solid and liquid lipids covered by a surfactant shell and are characterized by an inner disorganized network which allows a prolonged controlled release and a high drug loading capacity [12]. Over the past few years, growing studies have addressed the possibility of nanoencapsulation of phytochemicals in NLC systems. For instance, the synergism of herbal extracts-NLC (e.g., ivy leaves extract, marigold extract, willow bark extract) and their bioefficacy on antioxidant and antitumor activities have been reported by Lacatusu et al. [13,14,15,16]. Emerging trends in the development of advanced herbal active extracts-NLC based hydrogel evidence a strong anti-inflammatory effect [17,18] and improved anti-obesity action of endogenous lipid co-loaded lipid nanocarriers [19].

Recently, wild yam (*Dioscorea* spp.) has drawn scientists’ attention. It is recognized as a staple food in tropical and sub-tropical regions, also being considered a medicine [20]. It contains significant amounts of secondary metabolites, such as alkaloids, tannins, flavonoids, saponins (Diosgenin, Dioscin), glycoside steroids, anthraquinones, and polyphenols [21]. Owing to their structural complexity and synergistic biological activities, wild yam manifests distinctive properties, e.g., cell proliferation, anti-inflammatory potential, antifungal, anticancer, and hepatoprotective properties [22,23].

Diosgenin (DSG) is a natural plant bioactive, a steroid sapogenin consisting of a sugar molecule in combination with a triterpene or a steroid glycone [20]. It has an estrogenic structure, being used in hormone replacer therapy for middle aged women, with distinctive promise in menopausal disorders. Numerous studies have confirmed the influence of DSG in treating various inflammation processes, post-menopausal symptoms, and antispasmodic symptoms [24]. However, DSG therapeutic efficacy is compromised due to its poor bioavailability, improper permeability across lipid barriers, and extensive first-pass metabolism [25]. Its poor water solubility (0.02 mg/L) [26] and strong hydrophobicity [27] make it hard to be absorbed at optimal dose. Pharmacokinetic results showed an absolute low bioavailability of DSG in rats of only 6–7% [28]. Consequently, for keeping the beneficial characteristics and to avoid the previous mentioned shortcomings, DSG have been loaded in suitable delivery systems. Liu et al. thought to improve diosgenin solubility in water, to fully exploit the anticancer, estrogenic, and immunomodulatory effects, by capturing diosgenin in nanocrystals [27]. DSG loaded in niosomes have been developed to enhance the water solubility of diosgenin for better cancer prevention [24]. Liao et al. synthesized a conjugate of DSG with cytarabine to obtain self-assembled nanoparticles used in various bioactivities [29]. Ultimately, DSG polymeric nanoparticles or DSG nanocrystals showed aqueous solubility and superior oral bioavailability compared to DSG [28,30]. To our knowledge, no studies have reported on DSG encapsulated into nanostructured lipid carriers (NLC). Thus, the current study was undertaken to systematically develop new DSG–wild yam co-encapsulated NLCs to improve the oral bioavailability and biopharmaceutical properties, especially the antioxidant and anti-inflammatory ones, to naturally treat various health disorders. Although NLC are appropriate carriers for hydrophobic drugs, this study demonstrated the adaptability of these lipophilic delivery systems for effectively capturing the hydrophilic principles as wild yam extract (Yam).

## 2. Materials and Methods

### 2.1. Materials

A block copolymer by poly(ethylene glycol)-block-poly(propylene glycol) type (Poloxamer 188) and Polyoxyethylene sorbitan monolaureate (Tween 20/Tw20) were purchased from Sigma Aldrich Chemie GmbH (Munich, Germany). Glycerol monostearate (GMS) and Cetyl palmitate (CP) were purchased from Cognis GmbH (Monheim am Rhein, Germany) and Acros Organics (Morris Plains, NJ, USA) Sodium cholate (SC) from Merck (Darmstadt, Germany). Evening primrose (Oenothera biennis sp.) (EPO) and soybean (Glicyne max sp.) (SOY), vegetable oils, were ordered from Textron Plimon S.L.U. (Barcelona, Spain). The bioactive compounds, *Discorea villosa* extract (95% Diosgenin, DSG) (3β,25R)-(spirost-5-en-3-ol, C_27_H_42_O_3_) and wild yam extract (Yam) were supplied by Organic Herb Inc. (Changsha, China). Dimethyl sulfoxide (DMSO) and hydrogen peroxide were acquired from Merck (Darmstadt, Germany). Tris[hydroxymethyl] aminomethane, 5-Amino-2,3-dihydro-1,4-phthalazinedione (Luminol), 2,2-azinobis-(3-ethyl benzthizoline-6-sulfonic acid) (ABTS), 6-hydroxy-2,5,7,8-tetramethyl chroman-2-carboxylic acid (Trolox) were purchased also from Sigma Aldrich (Munich, Germany).

Cell lines and culture treatment: For the in vitro experiments, the endothelial HUVEC cell line isolated from the vascular endothelium of an umbilical cord was used (from American Type Culture Collection, ATCC). The treatment of adherent cells consisted in maintaining the culture in DMEM: F12 medium supplemented with 2 mM L-glutamine, 10% fetal bovine serum, 100 units/mL penicillin, 100 μg/mL streptomycin (Sigma Aldrich, St. Louis, MO, USA), and incubating at 37 °C in 5% CO_2_ humidified atmosphere.

### 2.2. Methods

#### 2.2.1. Synthesis of Nanostructured Lipid Carriers Loaded with DSG and/or Wild Yam

The nanostructured lipid carriers encapsulating DSG and/or wild yam extract (Yam) were obtained using high shear homogenization and high-pressure homogenization procedures applied successively to an o/w emulsion as described by Lacatușu et al. [19]. The lipid phase consisting in a blend of solid lipids (GMS, CP) and vegetable oils (EPO or SOY) and the aqueous phase (which contains 2.5% surfactant) were mixed under continuous stirring and were kept for 20 min at 73 °C. Proportions established between lipid matrix and herbal bioactive (DSG and/or wild yam extract) were chosen following the optimization step (Table 1). Afterwards, the pre-emulsion was submitted to the HSH (High Shear Homogenizer PRO250, Oxford, CT, USA) at 12,000 rpm for 1 min and HPH (APV 2000 Lab Homogenizer, Lubeck, Germany) for 6 homogenization cycles at 500 bars. The obtained nanodispersions were cooled at room temperature, stored overnight at −25 °C, and freeze dried by lyophilization (−55 °C, 54 h, using a Martin Christ Alpha 1–2 LD Freeze Dryer, Osterode am Harz, Germany).

#### 2.2.2. Particle Size Analysis

The size distribution of lipid nanocarriers, mean particle size (Z-ave), and polydispersity index (PdI) were analysed by dynamic light scattering (DLS) using a Zetasizer Nano ZS, Malvern Instruments Inc., Worcestershire, UK. After appropriate dilution of the aqueous NLC dispersion to adjust the scattering intensity, the measurements were achieved at a scattering angle of 90° and at room temperature. The particle size data were evaluated using intensity distribution. Each assay was performed in triplicate and data were calculated based on the Stokes–Einstein equation.

#### 2.2.3. Electrokinetic Potential Analysis

Zeta (electrokinetic) potential (ξ) determination (by means the measuring of the electrophoretic mobility in an electric field) was performed using the same Zetasizer Nano ZS, Malvern Instruments Inc., Worcestershire, UK. Prior to the analysis, NLC were dispersed in ultrapurified water and 0.9% sodium chloride solution, freshly prepared, to reach a conductivity of 50 µS/cm. Measurements were performed in triplicate and the results were expressed as mean ± S.D. Calculation of zeta potential involves the application of Helmoltz–Smoluchowski Equation (1):(1)$$\xi =\mathrm{EM}\;\frac{4\pi \eta }{\varepsilon }$$
where ξ is the zeta (electrokinetic) potential, EM electrophoretic mobility, η viscosity of the dispersion medium, and *ε* dielectric constant.

#### 2.2.4. Morphological Characterization

The morphological aspects of NLC-DSG-Yam were obtained using a Scanning Transmission Electron Microscope (Hitachi HD 2700, Hitachi High-Tech Corporation, Tokyo, Japan). In order to perform STEM analysis, the NLC-DSG-Yam-1/2 were appropriately diluted in bidistilled water and deposited on standard Cu TEM grids with Carbon thin layer film. Bright field mode was selected for capturing TEM images.

#### 2.2.5. Differential Scanning Calorimetry Analysis

The thermal behavior of the lyophilized-NLC was followed by differential scanning calorimetry (Netzsch Diferential Scanning Calorimeter 204F1 Phoenix, Selb, Germany). For each analysis, 20 mg of free and co-loaded NLC were weighed into aluminium pans and heated progressively from 25 to 100 °C, with a 10 °C/min cooling/heating rate, under constant nitrogen flow of 40 mL/min. As a reference, for getting the thermograms, an empty aluminium pan was used. The crystallinity index (CI) of NLC was calculated from the melting enthalpy (ΔH) and lipid concentration (c) using Equation (2):(2)$$\mathrm{CI}\left(\%\right)=\frac{{\Delta H}_{\mathrm{NLC}}}{{\Delta H}_{\mathrm{lipid}\;\mathrm{bulk}}\times {\;c}_{\mathrm{lipids}}}\times 100$$

#### 2.2.6. Encapsulation Efficiency (EE%) and Drug Loading (DL%)

Encapsulation efficiency of herbal bioactives (diosgenin and yam, individually and mixed), into NLC was determined indirectly by measuring the free bioactive compound (un-encapsulated bioactive) in the supernatant obtained after the centrifugation process. The HPLC method consists in weighing 0.05 g of lyophilized NLC-DSG, NLC-Yam, and NLC-DSG-Yam, addition of 1.5 mL of CH_3_OH, manual shaking and submission of suspension to a centrifugation process of 13,000 rpm, for 15 min, using a centrifuge (Sigma 2K15, Osterode am Harz, Germany). The resulted supernatant that contains the un-encapsulated DSG was subjected to HPLC determination (Jasco 2000 Liquid Chromatograph, Jasco, Tokio, Japan, Nucleosil C18 column, UV detector, λ = 203 nm). The mobile phase consisted of CH_3_CN: H_3_PO_4_ (0.5% *v*/*v*) and the flow rate was of 1 mL/min. The retention time was 13.54 min for DSG. The concentration of un-encapsulated herbal bioactive was calculated using the Equation (3) and the drug loading (%DL) was determined using the Equation (4):(3)$$\mathrm{EE}\left(\%\right)=\frac{W_{a}-{\;W}_{s}}{W_{a}}\times 100$$
(4)$$\;\%\;\mathrm{DL}=\frac{W_{a}-W_{s}}{W_{a}-W_{s}+W_{L}}\times 100$$
where Wa = the weight of herbal active (DSG or Yam) initially added in the NLC formulation; Ws = the analyzed weight of DSG/Yam in supernatant; W_L_ = the weight of lipids added into the NLC.

#### 2.2.7. In Vitro Antioxidant Activity of NLC against Long/Short-Life Free Radicals

##### Chemiluminescence Assay

The antioxidant activity of pure herbal bioactives, free-NLC and NLC-DSG-Yam was achieved by the chemiluminescence method (Chemiluminometer Turner Design TD 20/20, Sunnyvale, CA, USA). Cyclic hydrazide (luminol) was used as a light amplifying compound which emits light and is converted into an excited amino-phthalate dianion in the presence of oxidizing species. The free radicals generator system was formed by Luminol (10^−5^ M), H_2_O_2_ (10^−5^ M), and Tris–HCl buffer solution (pH = 8.6). The reference solutions were diluted after preparation in ethanolic solution. The free-radical-scavenger activity of the developed lipid nanocarriers was calculated by using the relation (5):(5)$$\mathrm{AA}\;\left(\%\right)=\frac{I_{0}-{\;I}_{s}}{I_{0}}\times 100$$

I_0_ = the maximum CL for reference at t = 5 s; I_s_ = the maximum CL for sample at t = 5 s.

##### ABTS Assay

For the evaluation of the antioxidant capacity of lipid nanocarriers against long-life radicals, by ABTS^●^^+^ type, a known spectrophotometric assay was used. The principle of this method is associated with the remaining long-life cation radicals that is reduced after the addition of an antioxidant in the reaction medium (λ = 734 nm). As ABTS^●^^+^ is reduced, the solution color changes. The ABTS^●^^+^ were generated from rection between ABTS solution (7 mM) and K_2_S_2_O_8_—water solution (2.45 mM) and kept in the dark at room temperature for 16 h before use. Sample preparation required the addition of 3 mL ABTS^●^^+^ solution to 1 mL NLC solution (0.05 g lyophilized-NLC into 10 mL ethanol) and 1 mL of ethanol. The blank solution was prepared identically, by replacing the volume of the NLC with the reference solution. Absorbance was measured using UV-VIS-NIR Spectrophotometer Type V670, (Jasco, Tokyo, Japan) at 4 min after mixing, using ethanol as reference. The scavenging capacity of ABTS^●^^+^ was calculated as inhibition (%) using Equation (6):(6)$$\;\%\mathrm{Inhibition}\;{\mathrm{ABTS}}^{●+}=\frac{A_{0}-A_{s}}{A_{0}}\times 100\;$$
where A_0_ = the absorbance of the blank (unscavenged radical cation solution); A_s_ = the absorbance after the addition of the NLC solution (antioxidant solution).

#### 2.2.8. In Vitro Controlled Release

The in vitro release experiments were performed by using an appropriate dialysis bag diffusion method (with regenerated cellulose bags). Briefly, 5 mL aqueous dispersion of NLC-DSG-Yam was placed into a dialysis bag (MWCO 12–14 kDa), which was then immersed in 50 mL of receiving medium (ethanol: phosphate buffer solution = 70:30, pH 6.4). The release environment temperature was kept at 37 °C and under continuous stirring, 300 rpm. At a fixed time, 1 mL of the samples was collected and replaced with fresh dissolution medium, maintained at the same temperature. The amounts of released herbal actives were analysed by the HPLC method considering the previously experimental procedure described in Section 2.2.6. The result was expressed as % of herbal active released (DSG, DSG + yam), compared to the amount of the initially added one, during the synthesis process.

#### 2.2.9. In Vitro Assessment of Cell Viability

##### Cytotoxicity by MTS Assay

MTS-based colorimetric assay was employed to evaluate the cell viability and cytotoxicity of developed lipid nanocarriers. This assay is based on cell’s ability to reduce the yellow tetrazolium salt (MTS) to the coloured formazan. The in vitro experiments were performed on a CellTiter 96 Aqueous One Solution Cell Proliferation Assay (Promega, Madison, WI, USA) in 96-well microtiter plates (Thermo Scientific, Waltham, MA, USA). Thus, 10^4^ cells/well were raised in 100 μL for 24 h, then culture supernatants were removed, and cells were treated with various concentrations of NLC-free and NLC-DSG-Yam-1 and 2, during 24 h or 48 h. After incubation, 20 μL of MTS and PES were added to each well. After colouring solution addition, plates were incubated for 4 h at 37 °C. The reduction of the tetrazolium salt was spectrophotometrically measured at λ = 492 nm (DYNEX Technologies MRS, Chantilly, VA, USA). The viability percentage was calculated compared to untreated cells (untreated cells served as control, being 100% viable) using the relation (7):(7)$$\mathrm{Cell}\;\mathrm{viability}\;\left(\%\right)=\frac{\left({\mathrm{Abs}}_{\mathrm{treated}\;\mathrm{cells}}-{\mathrm{Abs}}_{\mathrm{culture}\;\mathrm{medium}}\right)}{\left({\mathrm{Abs}}_{\mathrm{untreated}\;\mathrm{cells}}-{\mathrm{Abs}}_{\mathrm{culture}\;\mathrm{medium}}\right)\;}\;\times \;100$$

Results were expressed as mean values of three determinations ± standard deviation (SD).

##### Cytotoxicity by Real-Time Cell Analysis (RTCA)

Other quantification assays of cytotoxicity were achieved using xCELLigence technology and RTCA-DP analyzer (which use one to three 16 well E-plates). The RTCA-DP analyzer can automatically select probes to measure cell impedance and continuously transfer data (e.g., cell adhesion, cell morphology or cell viability) to the computer. To summarise, 1 × 10^4^ HUVEC cells were cultured in 100 μL culture medium in each well of E-Plates 16 cell (ACEA Biosciences, San Diego, CA, USA). Stabilization time for the cell culture was 10 min, afterwards plates were placed in the xCELLigence DP-System and into a 5% CO_2_ humidified incubator. Thus, xCELLigence System started recording the growth curves, in real time. Cell impedance changes were measured with the electrodes located at the bottom of each well and the data was continuously passed to the RTCA Software 1.1 (Mannheim, Germany) as cell index (CI) for further processing. After CI (cell index) value went to 1.0, different increasing concentrations of NLC-1/2, NLC-DSG-Yam-1/2 were used and living cells were closely followed.

#### 2.2.10. The Assessment of the Anti-Inflammatory Activity (ELISA Assay)

The anti-inflammatory action of NLC-DSG-Yam-1 and 2 was assessed using ELISA kits (‘‘Ready-Set-Go” Affimetrix eBioscience) determining the secreted proinflammatory cytokines level, IL-6 and TNF-alpha. Firstly, for capturing the specific antibodies an overnight incubation step, was required (4 °C). Then, after using a washing solution (PBS-Tween 20, 0.05%) and blocking the nonspecific sites for 60 min at room temperature (RT), the standards and compounds were added and reincubated in the previously conditions. Eighth concentration points served for the calibration curves, ranging from 500 pg/mL for IL-6 until 1000 pg/mL for TNF-6, after serial dilutions, 1:2. A room temperature incubation with specific detection antibodies was performed, after the washing step with PBS-Tween 20 solution. The specific molecules were detected by the addition of Avidin-HRP enzyme (30 min, RT) followed by TMB, substrate solution (15 min, RT). Prior to each of these two steps, several washes were performed. The NH_4_SO_4_ solution addition stopped the reaction and allowed optical densities to be measured spectrophotometrically at 450 nm. At the end of treatments of HUVEC cell culture supernatants, with NLC and/or H_2_O_2_, the IL-6 and TNF-α proinflammatory cytokines levels were evaluated. To perform the ELISA assay, the conditioned media were harvested, centrifuged for 30 min, and the supernatants were stored at −80 °C. The data obtained from the standard curves performed for each cytokine and experiment are expressed in pg/mL. The results were expressed by plotting the cytokine concentrations released against the untreated cell concentrations, considered 100%.

#### 2.2.11. Statistical Analysis

Data in all figures were expressed as the mean value of the measurements. Data were obtained in triplicate (n = 3), averaged and expressed as mean ± standard deviation (SD). For the in vitro experiments, data analyses were performed using GraphPad Prism 7 (GraphPad Software Inc., La Jolla, CA, USA). The differences between the treatment and control groups were statistically analysed using an unpaired two tailed t-test and one-way ANOVA. Statistical significance was considered at *p* < 0.05 and insignificance at *p* > 0.05 (NS).

## 3. Results and Discussion

### 3.1. Size and Stability Features of the NLC Loaded with Hydrophilic and Lipophilic Herbal Bioactive

The size and morphology of the lipid nanocarriers loaded with DSG and wild yam extract were provided by the ZC phase contrast micrographs (Figure 1). As shown by the DLS statistical analysis (Figure 2), the average diameters of the lipid nanocarriers containing DSG and Yam extract were between 50 nm and 120 nm, being well-defined nanospheres (Figure 1A). With the aid of co-localized STEM measurements and different magnifications, some internal structural nanoshapes were detected (Figure 1B). These small nanospheric inclusions less than 5 nm are easily observed in Figure 1B as uniformly dispersed in the lipid network of NLC and can be assigned to the herbal bioactive entrapped in the different compartments created by the vegtable oil used (EPO and SOY).

To assign the stability in time of the different categories of NLC (free, loaded with a single active principle, DSG or Yam extract, and those co-loaded with both types of plant bioactive principles), the mean diameter size evolution as well as the zeta potential characteristics were followed over a period of 60 days (Figure 2 and Figure 3). NLC that contain a single active principle, DSG or Yam, showed small changes 45 days after preparation. Some notable increases in Zave were determined for NLC-dual systems, which contain co-encapsulated DSG and Yam, but these increases do not compromise the effective stability of NLC-DSG-Yam. Based on the variation coefficients, e.g., 1.5–7.3% for the NLC-1 prepared with evening primrose oil and 3.3–8.9% for the NLC-2 prepared with soybean oil, it can be concluded that all the lipid nanocarriers manifest an appreciable physical stability, with this not being influenced by the type of herbal oil used in the preparation of the lipid core (Figure 2).

Overall, a slightly better physical stability of lipid nanocarriers with soybean oil/NLC-2 can be appreciated as compared to those of NLC prepared with evening primrose oil/NLC-1 (Figure 3). Encapsulation of DSG and Yam extract produced a slightly decrease in the zeta potential values. For instance, the determined zeta potentials for encapsulated NLC-1 were around (−40, −50) mV, while those for NLC-2 ranged between −44 and −54 mV. The last zeta potential values revealed that the encapsulation process has generated a higher electrical potential at the slipping plane in presence of soybean oil.

Zeta potential values evaluated for a period of two months have disclosed a predictable lack of coagulation over time of lipid nanoparticles. Moreover, the behaviour of NLC-DSG versus NLC-Yam or NLC-DSG-Yam was interesting. If in the case of NLC-DSG the ξ values were comparable in time, in the case of NLC which encapsulate Yam extract, a potentiation of the strong electronegative character was observed, after maintaining at 4 °C for 45 days. This result can be attributed to a re-organisation of the surfactant coating in which the hydrophilic herbal active (Yam extract) was entrapped with the exposure of many negative surface loads (Figure 3).

### 3.2. The Thermal Behaviour and Entrapment Characteristics of Lipid Nanocarriers Loaded with Herbal Actives

The scanning calorimetry results revealed three endothermic points corresponding to the melting process of constituent lipids. The individual pure lipids have a melting range of 57–65 °C for GMS and around 55–56 °C for PC. The presence of the three peaks in the bulk lipid mixture, with the slightly noticeable changes in melting points of pure lipids, points to the complexity of the lipid matrix, as well as the influence of unsaturated fatty acids from the herbal oils (EPO/SOY). These endothermic peaks from the bulk lipids were found in nanosized lipids/NLC in the form of a narrower melting range, much more obvious in the case of NLC prepared with soybean oil. The melting process of lipids from nanocarriers occurred at a lower temperature compared with the absolute lipids, due to lower particles size (Table 2). Same association between the smaller size after the nanosizing process and smaller melting points are reported by scientific studies [31]. According to classic thermodynamics, it is predictable that the melting points of solid lipids will decrease following the increase in mixture enthalpy [32]. Moreover, an additional explanation based on the Gibbs–Thompson effect states that the ratio between the specific surface area and particle size, with a smaller dimension, results in a lower melting point. 

DSC analysis of co-loaded NLC, of bulk lipids and free-NLC (Figure 4) pointed out the presence of two melting points for NLC-1 (with EPO), around 46.2 °C and 52.2 °C, respectively, with a principal melting process at 50 °C for NLC-2 (with SOY). For the co-loaded systems/NLC-DSG-Yam-1/2, these points were moved with about 1.5 °C, compared with the free-NLC. In addition, the peak enlargement reported in the case of NLC-DSG-Yam is evidence of lipid core disturbance after loading with DSG. Widening of the endothermic curve is proof of a messier and more distorted lipid network.

One other important aspect is linked to the influence of the herbal oil (EPO or SOY) in building the inner lipid network of NLC. The presence of SOY has generated a shrinking of the melting specific area for the nanocarrier with soybean oil, suggesting the presence of a more ordered network (Figure 4b). This assumption can also be materialized by the values of crystallinity indices calculated from ΔHm: 27.7% and 34.76% (for NLC-free-1 and NLC-DSG-Yam-1) versus 44.8% and 39.6% (for NLC-free-2 and NLC-DSG-Yam-2).

The ability of nanocarriers to entrap DSG and Yam, by quantifying the entrapment efficiency and loading capacity showed the remarkable ability of nanocarriers to capture both kinds of herbal bioactives with different polarities or affinities (Figure 5). Diosgenin was rather preferentially attired by the lipid core while the yam extract captured by the shell formed from surfactant or co-surfactant mixture: EE_Yam_ = 84.3% ± 3.38% for NLC-Yam-1 and 83% ± 0.51% for NLC-Yam-2; EE_DSG_ = 86.9% ± 1.58% for NLC-DSG-1 and 85.7% ± 1.26% for NLC-DSG-2. A small decrease in encapsulation efficiency was reported for NLC-DSG-Yam, but values remained at >80%. For instance, EE_yam + DSG_ = 84% ± 3.54% for NLC-DSG-Yam-1 and 81.2% ± 3.93% for NLC-DSG-Yam-2 (Figure 5a).

The determination of LD% (which considers the weight of lipids added in the NLC) highlighted that the DSG prefers the lipophilic lipid core, while Yam extract, of hydrophilic nature, does not significantly impact the LD value (Figure 5b). This aspect strengthens the preferential distribution of the hydrophilic extract in the surfactants coating of NLC.

### 3.3. Assigning In Vitro Cytotoxicity to NLC Systems

The NLC behaviour on the cell viability of HUVEC endothelial cells, shown in Figure 6, highlights the impact of NLC concentration, especially at concentrations higher than 100 μg/mL on these cell types. At these concentrations, the viability is quite low but maintaining the treatment in a range between 3 and 50 μg/mL ensures a cell viability that is maintained at values > 70%. These results indicate a low cytotoxicity effect of developed NLC-DSG-Yam.

By continuing the treatment for another 24 h (Figure 6b) the cell viability was strongly improved. This prolonged NLC-DSG-Yam treatment on HUVEC endothelial cells (48 h) led to a counterbalance of cell viability as compared to the results obtained after 24 h of treatment: values higher that 85% in 50–3.125 μg/mL were detected for NLC-DSG-Yam-2. The increase in endothelial cell survival at prolonged treatment could be assigned to cell proliferation. This proliferation effect could be a result of the low toxicity of NLC, which allows these endothelial cells to further proliferate. Based on these observations, NLC-DSG-Yam-2 can be classified as non-toxic towards endothelial normal cells.

The results obtained by the RTCA assay on the HUVEC endothelial cells sustain the data from the previous colorimetric MTS analysis. Following the comparison between the cytotoxicity and cell proliferation of HUVEC cells, at different concentrations of NLC, it could be observed that concentrations below 100 μg/mL assure a desired cell viability, comparable with the non-treated cells (red curve). Interestingly, as the treatment progresses, for concentrations between 25 and 50 μg/mL, the cell viability is almost similar to those of non-treated cells, indicating a complete lack of cytotoxic effect (Figure 7).

### 3.4. In Vitro Antioxidant Activity through TEAC and Chemiluminescence Method

The antioxidant activity of the developed nanocarriers was determined through two different methods, chemiluminescence for capturing short-life free oxygenated radicals, respectively ABTS, and quantifying the inhibition of long-life stable, cation radicals.

Following the analysis of lipid nanocarriers, free- and co-loaded as for the bioactive herbal principles, it has been observed that they possess a low capacity of scavenging these cation radicals (e.g., a modest value of 12.4% was obtained for the NLC-DSG-Yam-1). Even though all the systems have been reduced at a small size scale, the antioxidant activity has an increase of only 1.5%.

The chemiluminescence assay has highlighted that the antioxidant ability of NLC is somewhat more effective in the case of short-lived oxygenated free radicals than for cationic ABTS radicals. Both vegetable oils, EPO or SOY, and DSG and Yam themselves have pretty good antioxidant activity. Regarding the free-NLC and NLC-DSG or NLC-Yam, they are not so considerably different in terms of antioxidant activity, with the values varying between 71% and 79%. ω-3 and ω-6 fatty acids from EPO and SOY could be responsible for these results. However, a noticeable improvement can be observed in the nanocarrier co-loaded with DSG and Yam, both with evening primrose as with soybean oils, 82% ± 0.74% for NLC-DSG-Yam-2 and 84% ± 1.6% for NLC-DSG-Yam-1 (Figure 8b).

### 3.5. In Vitro Controlled Release

Experiments designed to emphasize the release behaviour of bioactive herbal co-opted into the same lipid nanocarrier delivery system prepared with evening primrose oil or soybean oil revealed a great difference between NLC loaded with an individual herbal compound and the NLC co-loaded with DSG and yam extract. An unaltered surfactant coating, composed of SC and Tw 20, favoured the short-term release of DSG, most likely by the rapid dissolution of the NLC coating. For example, DSG was released in a percentage of almost 100% in 8 h, independent of the type of vegetable oil, from the individual NLC systems. In contrast, an altered NLC coating in which the wild yam extract (hydrophilic in nature) is accommodated led to a delay or impairment of DSG release from the lipid core of NLC (Figure 9). For the co-loaded NLC, the release process was slower and more sustained during a period of 24h. The cumulative release percentage reached 88.02% ± 1.68% in the case of NLC-DSG-Yam-1 and 85.5% ± 0.3% for NLC-DSG-Yam-2.

### 3.6. In Vitro Anti-Inflammatory Effect of NLC-DSG-Yam

The natural bioactives can be used as inhibitors of pro-inflammatory cytokines to treat inflammatory conditions. The encapsulation of herbal actives that contain different biologically active compounds into lipid nanocarriers is an important premise in obtaining improved therapeutic effects, mainly determined by their more effective access to the site of inflammation and the synergism that occurs by combining several types of bioactive principles present in the same delivery system. In this part of study, the levels of IL-6 and TNF-α were measured to assign the potential anti-inflammatory effect of the developed NLC-DSG-Yam.

Evaluation of the pro-inflammatory expression (ELISA assay, Figure 10) revealed an enhanced anti-inflammatory effect. By treating HUVEC cells with NLC, the production of pro-inflammatory cytokines TNF-α and IL-6 was significantly inhibited (Figure 10). The inhibition study of the cytokines pointed out a strong inhibition dependent on the treatment applied NLC dose and the encapsulated bioactive. A dose of 50 µg/mL of NLC exhibited a more accentuated inhibition of IL-6 and TNF-alpha cytokines compared to the higher one, 200 µg/mL. This more effective counteracting of the lower dose of NLC respectively 50 µg/mL, compared to the treatment with a higher concentration, 200 µg/mL, can be explained according to the MTS and RTCA results which demonstrated a decrease in cell viability at concentrations of 200 µg/mL. This behaviour results in a denaturation of cells, or even death, which directly impact the inhibition of pro-inflammatory cytokines.

Furthermore, a clear distinction, in terms of inhibition percentage, can be made between the free- and DSG-Yam co-loaded NLC. Another visible aspect is the faster inhibition of TNF-α, compared to IL-6, for all analysed NLC having a TNF-α inhibition higher that 80% at a specified concentration of 50 µg/mL. These results may be associated with the action of the oxidizing agent (H_2_O_2_) applied to HUVEC cells. Hydrogen peroxide resulted in a higher release of TNF-α compared to IL-6. In addition, by comparing the two types of NLC that contain EPO or SOY, it could be noticed that the inhibition of the two cytokines is more accentuated in the case of NLC-2/with SOY than for NLC-1/with EPO. The most relevant example is the cytokine inhibition of NLC co-loaded with DSG and Yam which yields an inhibition of 98.2% ± 1.07% of TNF-α for NLC-DSG-Yam-2 and of 90.42% ± 1.17% for NLC-DSG-Yam-1, respectively.

## 4. Conclusions

In this study, negatively charged lipid nanocarriers loaded with two kinds of lipophilic (Diosgenin) and hydrophilic (wild yam extract) herbal bioactive were fabricated. The encapsulation process of the two bioactive principles, as well as the physical stability and mean particle size, provided sufficient evidence for the long-term stability, as for the encapsulation efficiency for the intended purpose. The performance efficiency of the newly developed dual herbal-loaded NLC (NLC-DSG-Yam) was tested against HUVEC endothelial cells, via an in vitro cell cytotoxicity assay (MTS and RTCA) and anti-inflammatory action (ELISA test), in vitro release evaluation, and in vitro antioxidant capacity (chemiluminescence and TEAC assays).

The results obtained for the antioxidant activity revealed the direct influence of the bioactive compound and the type of vegetable oil employed for preparation of NLC. The co-loaded NLC assured a much slower DSG release during a period of 24 h as compared with the NLC encapsulating only diosgenin and a more emphasized ability of capturing short-life radicals than cationic stable radicals. Although these nanocarriers own moderate capacity of catching the cation radicals ABTS^+^, through chemiluminescence assay NLC-DSG-Yam manifested an improved percentage of 82% to 84% to capture free oxygenated radicals. NLC safety properties monitored following the in vitro MTS and RTCA assays demonstrated that concentrations less than 50 μg/mL showed no cytotoxic effects during in vitro cytotoxicity assays. Moreover, concentrations of 50 µg/mL exhibited a more accentuated inhibition of IL-6 and TNT-α cytokines, compared to the highest one, 200 µg/mL. The co-loading of DSG and wild yam extract led to an amplified anti-inflammatory effect towards TNF-α, e.g., 98.2% ± 1.07% for NLC-DSG-Yam-2 and 90.42% ± 1.17% for NLC-DSG-Yam-1. Hereby, we conclude that the herbal bioactives encapsulated by nanocarriers will play an increasingly promising role in future therapeutics, seeing that the appropriate use of lipid nanocarriers enhances several desirable properties of herbal bioactives.

## Figures and Tables

**Figure 1 nanomaterials-11-03035-f001:**
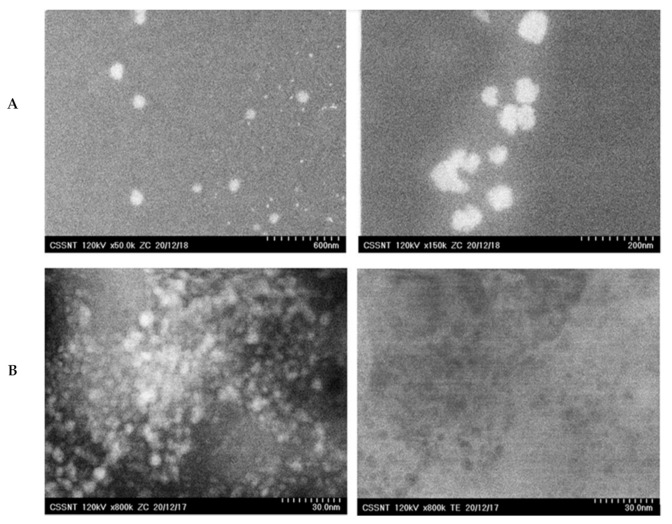
TEM images of NLC-DSG-Yam (**A**) Nanosphere diameter measurements. The diameters of the nanosphere were measured in ZC and TEM micrographs at different magnifications: ×50 K and ×150 K. (**B**) Co-localized measurements. Images obtained using different detectors (left, ZC—phase contrast and right—TE—transmission electrons), same area of the sample, at the same magnification.

**Figure 2 nanomaterials-11-03035-f002:**
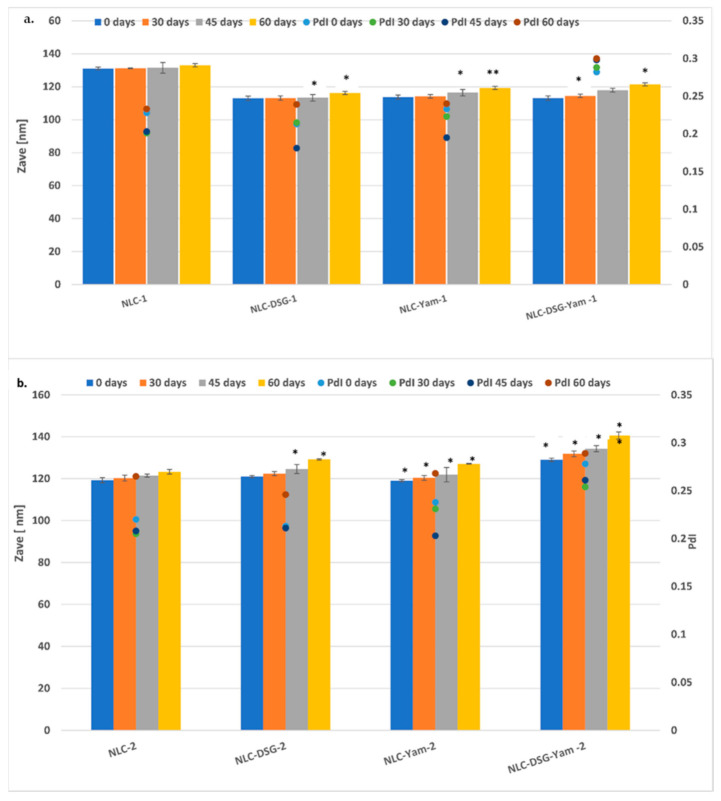
Variation of mean diameters (Zave) and polydispersity index (PdI) for the nanocarriers prepared with EPO (**a**) and for those prepared with SOY (**b**). * *p* < 0.05; ** *p* < 0.005; NS *p* > 0.05 (NS). Data are expressed as mean ± SD, n = 3 NLC-DSG1/2, 0 days vs. other groups.

**Figure 3 nanomaterials-11-03035-f003:**
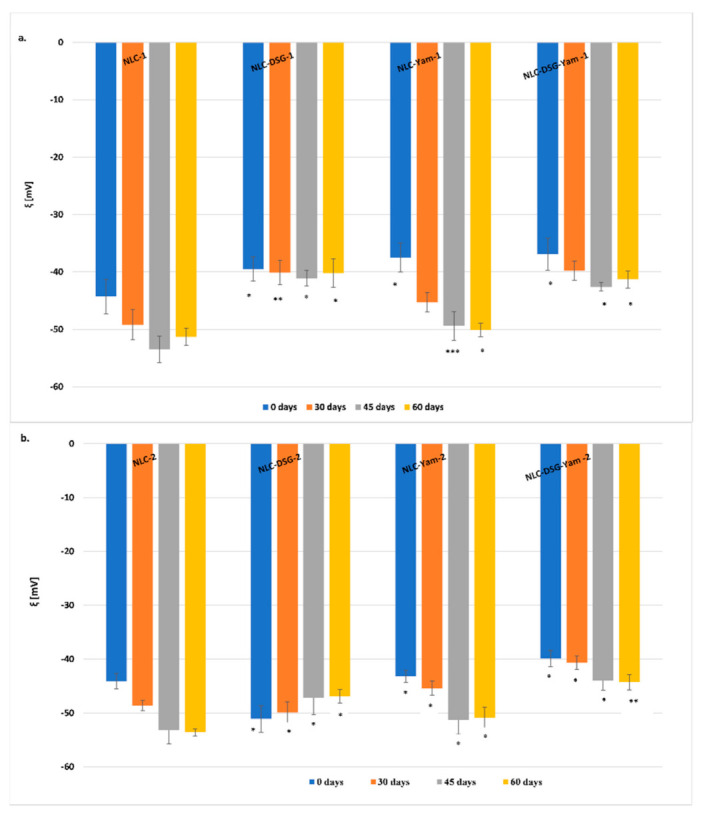
Variation over time in the physical stability of aqueous free- and co-loaded-NLC: (**a**) NLC-1 (prepared with evening primrose oil); (**b**) NLC-2 (prepared with soybean oil). * *p* < 0.05; ** *p* < 0.005; NS *** *p*< 0.0005; *p* > 0.05 (NS). Data are expressed as mean ± SD, n = 3, NLC-DSG1/2, 0 days vs. other groups.

**Figure 4 nanomaterials-11-03035-f004:**
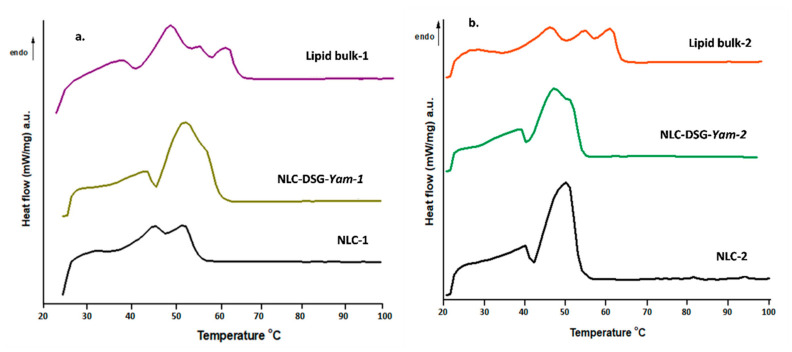
Comparison between DSC curves of bulk lipids, free-nanocarriers and nanocarriers loaded with DSG and *Yam*, with EPO (**a**) and SOY (**b**).

**Figure 5 nanomaterials-11-03035-f005:**
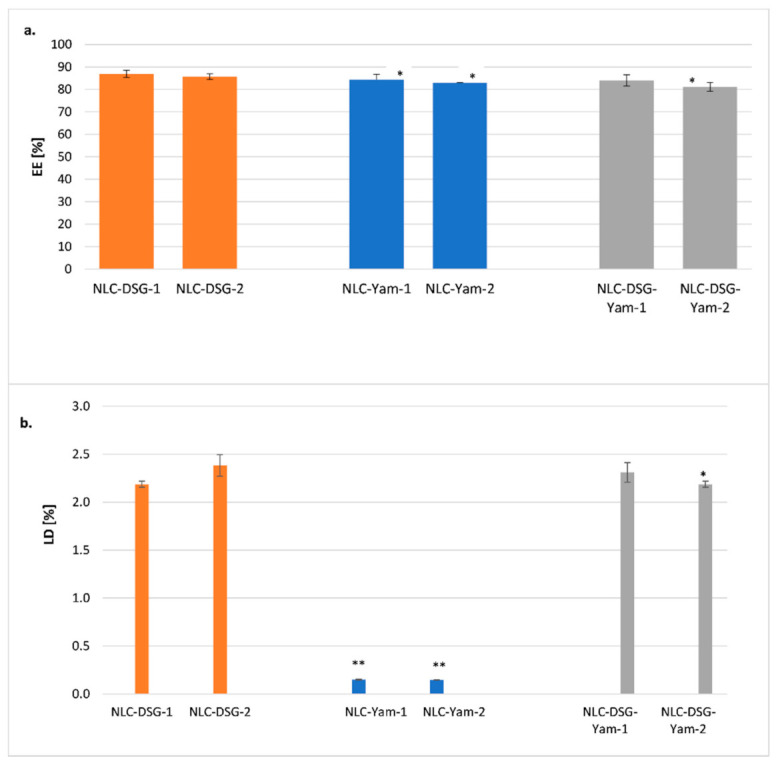
Encapsulation efficiency (**a**) and loading capacity (**b**) of DSG and *Yam.* * *p* < 0.05; ** *p* < 0.005; NS *p* > 0.05 (NS). Data are expressed as mean ± SD, n = 3 NLC-DSG1/2 vs. other groups.

**Figure 6 nanomaterials-11-03035-f006:**
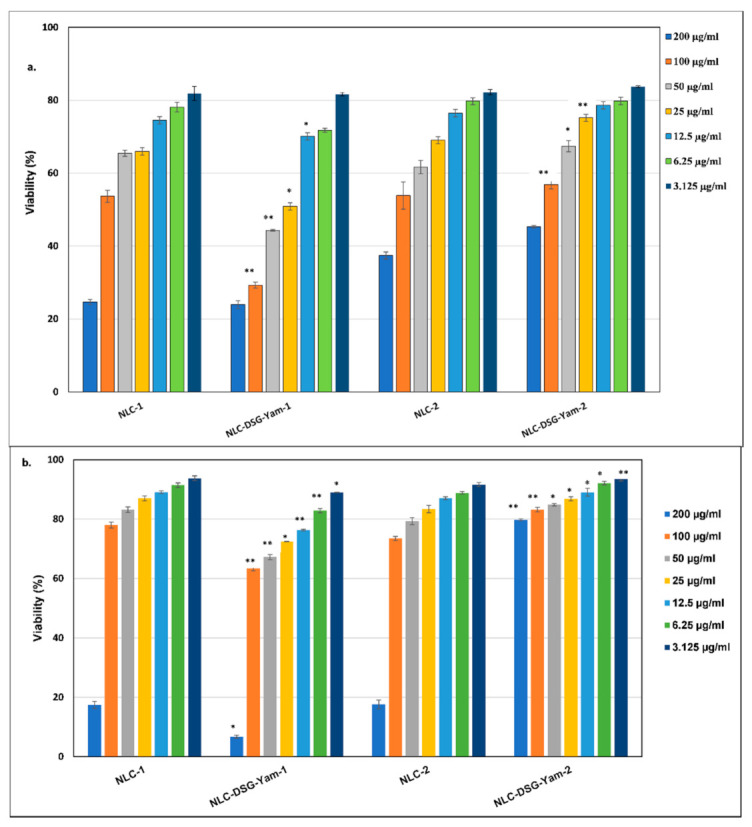
The effect of NLC-DSG-*Yam* as compared to NLC-free, on cell viability of HUVEC endothelial cells, after 24 h (**a**) and 48 h of treatment (**b**) * *p* < 0.05; ** *p* < 0.005; NS *p* > 0.05 (NS). Data are expressed as mean ± SD, n = 3 NLC-1/2 vs. other groups.

**Figure 7 nanomaterials-11-03035-f007:**
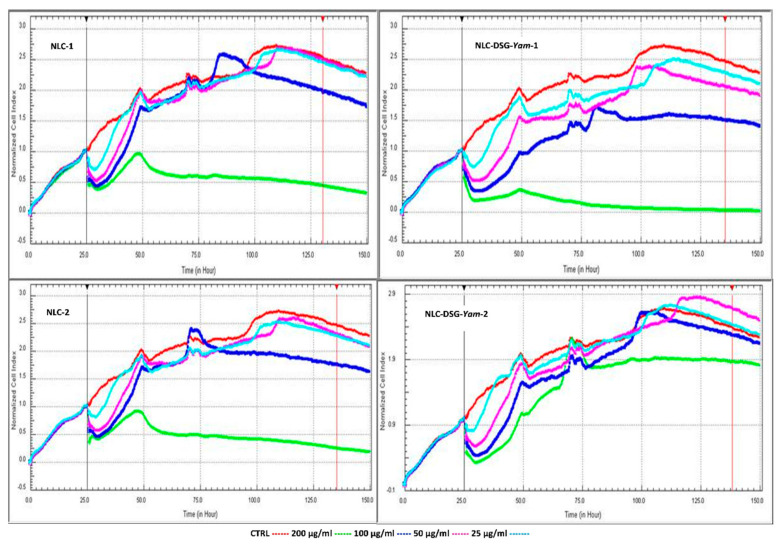
Cytotoxic vs. proliferation induced by NLC-DSG-Yam as compared with free NLC.

**Figure 8 nanomaterials-11-03035-f008:**
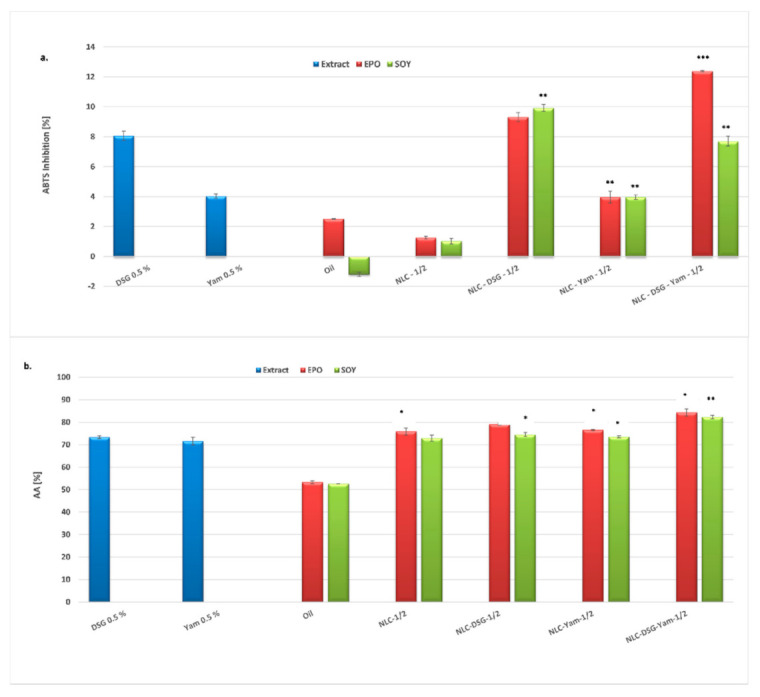
Antioxidant ability of NLC-DSG-Yam, determined by: A. TEAC assay (**a**); B. chemiluminescence method (**b**) * *p* < 0.05; ** *p* < 0.005; *** *p*< 0.0005; NS *p* > 0.05 (NS). Data are expressed as mean ± SD, n = 3 NLC-1/2 vs. other groups.

**Figure 9 nanomaterials-11-03035-f009:**
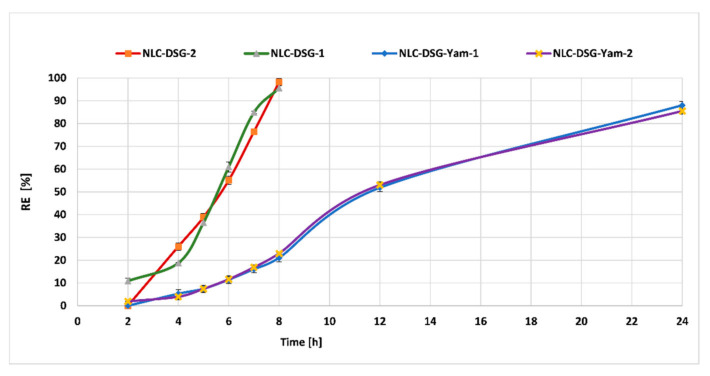
In vitro release of the DSG from NLC-DSG and NLC-DSG-*Yam*.

**Figure 10 nanomaterials-11-03035-f010:**
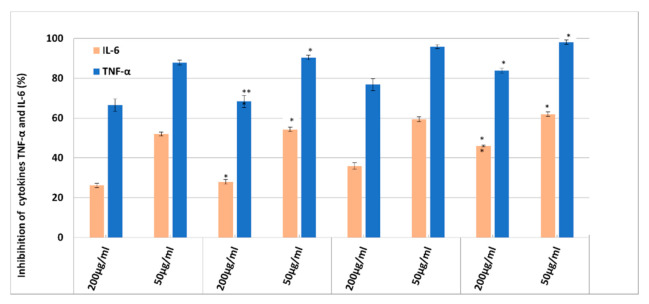
Inhibition action NLC-DSG-*Yam* on the release of TNF-α and IL-6 pro-inflammatory cytokines. * *p* < 0.05; ** *p* < 0.005; NS *p* > 0.05. Data are expressed as mean ± SD, n = 3 NLC-1/2 vs. other groups.

**Table 1 nanomaterials-11-03035-t001:** Composition of nanostructured aqueous dispersions encapsulating Diosgenin and/or wild *yam* extract.

Type of NLC *	Lipid Phase (10%), g			Surfactant Mixture (2.5%) g			Herbal Bioactives ***, g	
	GMS	CP	EPO/SOY	Polox 188	Tw 20	SC	DSG	Yam
NLC-1/2 **	3.5	3.5	3.0	0.375	0.875	1.25	-	-
NLC-DSG-1/2							0.5	-
NLC-Yam-1/2							-	0.5
NLC-DSG-Yam-1/2							0.5	0.5

* The amount up to 100% is represented by distilled water; ** NLC-1 were obtained by using evening primrose oil/EPO, while NLC-2 were obtained by using soybean oil/SOY; *** DSG has been added to the melted lipid phase, while the wild yam extract was solubilized in the aqueous surfactants phase.

**Table 2 nanomaterials-11-03035-t002:** Characteristics of bulk lipids, free and DSG and/or *Yam* co-loaded lipid nanocarriers.

Sample	Tm (°C)	ΔHm (J/g)	IC	Tc (°C)	ΔHc (J/g)
NLC-1	46.2	6.65	-	35.8	28.7
	52.2	51.08	27.67		
NLC-DSG-Yam-1	44.2	9.46	-	37.4	21.58
	53.1	64.17	34.76	42	2.935
Lipids bulk 1 (MSG + PC + EPO)	37.4	7.61	100	37.9	16.26
	48.6	18.46		41.5	1.98
	55.9	1.84		47.3	8.08
	61.7	9.01		-	-
NLC-2	39.9	8.59	-	38.5	25.02
	50	58.66	44.81	43.2	4.736
NLC-DSG-Yam-2	38.8	7.232	-	38	15.75
	48.5	51.84	39.60	42.7	3.288
Lipids bulk 2 (MSG + PC + SOY)	47.1	13.09	100	38.8	15.82
	55.5	3.8		42.4	2.8
	62	9.51		48.1	8.169

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
