# Peer review of "Challenges in Coopted Hydrophilic and Lipophilic Herbal Bioactives in the Same Nanostructured Carriers for Effective Bioavailability and Anti-Inflammatory Action"

_nanomaterials, 2021, doi:10.3390/nano11113035_

Round 1
Reviewer 1 Report
- Why does the average particle size become smaller when NLC-1 is loaded with DSG, Yam or both?
- What is the difference between NLC-2 and NLC-1? Why does it show different load capacity?
- At a concentration of 50μg/ml for 24h, the survival rate of the cells is only about 70%, why is it considered that the concentration is lower than 100μg/ml is beneficial?
- Under high concentration conditions (200μg/ml), the cell survival rate after incubation for 24 hours is already lower than 50%. How to explain the cell survival rate increase after 48 hours incubation?
- The ROS removal experiment did not show the advantages of nano-loading, and the result only showed the superimposed effect of the two drugs.
- Some formulas are written incorrectly, and the format of references is not uniform.
Author Response
Dear Reviewer,
Based on your comments, we have carefully reviewed the manuscript.
Thank you for the valuable suggestions which help us in improving the research entitled “Challenges in coopted hydrophilic and lipophilic herbal bioactives in the same nanostructured carriers for effective bioavailability and anti-inflammatory action”.
All suggestions and observations provided by the Reviewer have been addressed and presented in the revised manuscript with a yellow background.
Sincerly yours,
Prof. Ioana Lacatusu
[email protected]; [email protected]
University POLYTHECHNICA Bucharest,
Faculty of Applied Chemistry and Material Sciences

Reviewer 2 Report
In this study, the authors emphasized that DSG was encapsulated with NLCs to improve the shortcomings of DSG, solubility and bioavailability. Therefore, NLC-DSG-yam extract was developed to improve the efficacy of herbal digen(DSG) in the treatment of anti-inflammatory diseases, and achieved the expected stability and encapsulation efficiency.However, there are still some issues need to be revised. This manuscript can only be accepted in the Nanomaterials after the following issues being well addressed:
1.The tenth line of the summary is misspelled, "ant" should be "and"
2.The resolution of Figure 1 is not clear enough, and it will become more blurred after magnification.
3.The experimental part of 3.3 Assigning in vitro cytotoxicity to NLC systems can be appropriately added with some experimental diagrams instead of simple chart data, which is not convincing.
- More ref to be cied: DOI: 10.1021/acsabm.9b00843 ACS Appl. Bio Mater. 2020, 3, 86−106; Tumor Microenvironment-Specific Functional Nanomaterials for Biomedical Applications;Journal of Biomedical Nanotechnology Vol. 16, 1325–1358, 2020.
Author Response

(The authors gave the same response as above.)
